# Opportunities and challenges in access to healthcare for international migrants with work-related diseases and injuries in Gulf Cooperation Council countries: A systematic literature review protocol

Weijun Yu[1]*, Aleena Dawer[2], Jeanetta Floyd[3], Nicole Saad[3], Jiaqin Wu[1], Katherine O. Robsky[1], Oliver Johnson[2], Yulia Hutsul[4], Dylan Ratnarajah[5], Bryan Shaw[1], Martine Etienne-Mesubi[1], Deus Bazira[1]*

1 Center for Global Health Practice and Impact, Georgetown University, Washington, DC, United States of America, 2 Global Health Institute, Georgetown University, Washington, DC, United States of America, 3 College of Arts and Sciences, Georgetown University, Washington, DC, United States of America, 4 Walsh School of Foreign Service, Georgetown University, Washington, DC, United States of America, 5 School of Medicine, Georgetown University, Washington, DC, United States of America

* weijun.yu@georgetown.edu (WY); db1432@georgetown.edu (DB)

## Abstract

### Introduction

International labor migrants form a significant part of the global workforce, particularly in the Gulf Cooperation Council (GCC) countries, which host around 11% of the world's migrant workforce. This high concentration presents unique challenges in healthcare access and delivery. This systematic review aims to evaluate whether international labor migrants in GCC countries have effective access to healthcare for work-related diseases and injuries and to propose evidence-based recommendations for policy and healthcare interventions.

### Methods

We will include studies from 2013 to 2023 published in peer-reviewed journals in English or Arabic (with English abstracts) available on PubMed, Embase and CINAHL. Search strategies are developed using MeSH terms and key terms related to our study population (international labor immigrants), context (the GCC countries), and exposure (migrant status; work-related diseases and injuries). The screening process involves two stages: initial review of titles/abstracts and full-text review. Studies meeting eligibility criteria and focusing on our primary outcome (access to healthcare) will be included. Data extraction will cover study characteristics, population demographics, described exposures, outcomes measured, and key findings.

**Data availability statement:** Data will be made available upon the completion of the full systematic review.

**Funding:** The author(s) received no specific funding for this work.

**Competing interests:** The authors have declared that no competing interests exist.

Given the expected heterogeneity, narrative synthesis will be primarily used, with meta-analysis as an option.

## Discussion

By considering both migrant workers and expatriate professionals, we provide a culturally tailored perspective. Methodological rigor is ensured through the gold standard screening process, where at least two reviewers independently screen the literature at each stage, with a senior reviewer resolving discrepancies. We will identify barriers, facilitators, and inform targeted interventions for policymakers. Our findings will support evidence-based strategies to improve healthcare access for international labor migrants in the GCC countries.

## Systematic review registration

This systematic review protocol was registered on the international registry PROSPERO (CRD42024532851) on April 21, 2024.

## Introduction

International labor migrants, encompassing both international migrant workers and expatriate professionals, form a critical segment of the global workforce. International migrant workers are individuals aged 15 and over, who relocate from their country of origin to participate in the labor force of their host country. Their work supports essential industries, and services across the globe, significantly contributing to economies, particularly in regions with high reliance on migrant labor [1]. Expatriate professionals, on the other hand, are highly skilled individuals employed for their specialized skills, abilities, and experience, engaging in global careers [2,3]. Regardless of their essential roles, the health necessities of international labor migrants are frequently overlooked [4].

This under-addressed issue is particularly important because international migrant workers are often employed in low-wage, high-risk jobs with exposure to hazardous conditions. Globally, over 150 million migrant workers experience circumstances associated with an elevated prevalence of occupational morbidity and mortality [5,6]. International migrant workers encounter significant barriers to healthcare access in their host countries including limited or no health insurance, restricted rights to statutory health care [7,8], language barriers, and unfamiliarity with local healthcare infrastructure [9]. A previous systematic review and meta-analysis study covering Asia, the Middle East, Europe and the Americas shows that globally, about 47% of international migrant workers suffer from occupational health issues, and 22% report workplace injuries or accidents [10]. The International Labour Organization (ILO) reports that approximately 7600 people die daily from work-related injuries or illnesses, with 15% of deaths directly attributable to work-related injuries [11]. Despite calls from international organizations like the World Health Organization for governments and

policymakers worldwide to adopt strategies that enhance occupational health for international migrant workers, the implementation of such strategies remains limited [7,12–14].

The Gulf Cooperation Council [15] countries (Bahrain, Kuwait, Oman, Qatar, Saudi Arabia, and the United Arab Emirates) constitute the largest hub for international labor migrants, hosting nearly 30 million migrants [16], which accounts for around 10% of the international migrant workforce [17]. Among these countries, Qatar, Kuwait, and the United Arab Emirates (UAE) have the highest percentage of migrants [16,18]. This significant influx of labor migrants presents substantial healthcare challenges, exacerbated by the transient nature of migration in the region. In Qatar, labor migrants have been regulated through the kafala (sponsorship) system, which legally binds employees to their employers [19]. Initially intended to facilitate labor migration and oversight [20], the system has raised concerns for restrictive practices and power imbalances. According to a 2021 investigation, approximately 6,500 migrant workers died in Qatar between 2010 and 2020, including deaths from all causes [21].

In many labor-intensive sectors such as construction and agriculture, international migrant workers are predominantly represented and frequently exposed to extreme heat conditions, which has been linked to severe health outcomes including cardiac arrest and heat exhaustion [20,22]. A significant number of these migrant workers originate from Southeast Asia; notably, nearly 34,000 Indian migrant workers reported dead in the GCC countries between 2014 and 2019 attributed to extreme working conditions [23]. The Covid-19 pandemic further underscored the vulnerability of these migrant workers, shifting the risk from merely physical injuries to now infectious diseases [7,24]. Concurrently, the rapid economic development in the GCC countries has led to an increase in work-related injuries among the expatriate professionals, contributing significantly to non-fatal injuries [25]. While each GCC country has its own healthcare regulations concerning labor force, the pandemic exacerbated the situation, leaving many individuals vulnerable, without access to essential medical service, or unable to return home for care [26–28].

From a patient-centered perspective, access to healthcare is defined as the ability to identify, seek, reach, obtain, and utilize healthcare services in a manner that addresses an individual's healthcare needs [29]. This concept integrates multiple dimensions of service acquisition and delivery, including availability, appropriateness, preference, and timeliness, to provide effective care for patients [30]. Achieving satisfactory access to healthcare is characterized by several factors: having health insurance that simplifies integration into the healthcare system, being able to seek necessary services promptly, or having a consistent healthcare provider to build a strong patient-provider relationship [31].

Given the critical contribution of international labor migrants to the economic development of the GCC countries, it is crucial to examine the healthcare opportunities and challenges specific to this population. While individual studies have explored these issues, at the time of writing, no systematic review has investigated international labor migrants' access to healthcare for work related conditions. Bridging this gap is essential for supporting targeted healthcare policy and practice.

## Objectives

The objective of this systematic review is to comprehensively summarize findings on whether international labor migrants experiencing work-related diseases and injuries in the GCC countries have efficient and effective access to healthcare services. Our goal is to provide evidence – based insights into the barriers and facilitators that may influence healthcare access and to propose recommendations that will inform the development of policies and healthcare interventions aimed at enhancing access to healthcare services for international labor migrants facing work-related diseases and injuries in the GCC countries.

## Research questions

Our research question is structured following the Population, Exposure, Outcome (PEO) framework. We consider international labor migrants in the GCC countries as our population, with migrant status and their experience of work-related diseases and injuries as our exposures, and access to healthcare services as our outcome. Our main research question is

"Whether international labor migrants experiencing work-related diseases and injuries in the GCC countries have access to healthcare services, and what factors influence this access over the past decade?"

Additionally, this review will explore the following subtopics:

a. Barriers and facilitators affecting timely access to healthcare services

b. Policies developed and implemented to address constraints on healthcare access for international migrants

c. Scope and coverage of healthcare services available to this population

d. Affordability of services and healthcare financing mechanisms

e. Patterns of healthcare service utilization

f. Health workforce compatibility, including language barriers, cultural competence, and availability of migrant-friendly services

g. Effectiveness of healthcare service, including quality, responsiveness, and overall impact

## Methods

### Study design

This systematic review protocol was developed in accordance with the *Preferred Reporting Items for Systematic Reviews and Meta-Analyses Protocols (PRISMA-P) Checklist* [32], which is the most current guideline for the development of systematic review protocols. Our PRISMA-P checklist is provided in S1 Appendix. On April 21st, 2024, our protocol was successfully registered on the International Prospective Register of Systematic Review (PROSPERO) with the registration number CRD42024532851 [33]. The full systematic review findings will be reported by following the *Preferred Reporting Items for Systematic Reviews and Meta-Analyses (PRISMA) 2020 guidelines* [34], which are used for reporting completed systematic reviews. A PRISMA flow diagram will be created to describe our process of identification, screening, eligibility, and inclusion of the final systematic review. This study does not involve human subjects; therefore, ethical review was formally waived by Georgetown University's Institutional Review Board committee (IRB ID: STUDY00007743).

### Eligibility criteria

The eligibility criteria is outlined in Table 1. We will include peer-reviewed articles written in English or Arabic (with English abstracts available) published between January 1st, 2013 and December 31st, 2023. The focus will be access to health-care for work-related diseases and injuries or occupational health among international labor migrants in the Gulf Cooperation Council countries (Bahrain, Kuwait, Oman, Qatar, Saudi Arabia, and the United Arab Emirates).

### Information sources

Three databases will be searched for eligible literature: PubMed, EMBASE, and CINAHL. Full texts of articles will be retrieved if they meet the eligibility criteria based on titles, abstracts, and key descriptors. Additionally, we will screen the reference lists and bibliographies of included articles to identify relevant articles that may have been missed during the initial screening. RefWorks software will be used to manage references and remove duplicates.

### Search strategy

Search terms, keywords, Medical Subjects Headings (MeSH) terms, and Boolean operators will be used to develop search strategy for each database. We formulated our search algorithm based on terms related to our study population (international immigrants), context (the GCC countries), and exposure (migrant status; work-related diseases and injuries),

**Table 1. Eligibility Criteria.**

| Criteria | Inclusion | Exclusion |
|---|---|---|
| **Population** | - International migrant workers moved to GCC (Gulf Cooperation Council) countries (Bahrain, Kuwait, Oman, Qatar, Saudi Arabia, and the United Arab Emirates) for labor-intensive employment opportunities, driven by economic necessity. <br> - Expatriate professionals relocate for specialized or managerial roles with competitive benefits. | - International migrants living outside GCC countries. <br> - Local migrants with GCC citizenship status. |
| **Language** | - Literature written in English. <br> - Literature written in Arabic with English abstracts available. | Literature written in other languages without English abstracts. |
| **Time Frame** | Studies conducted between January 1st, 2013 and December 31st, 2023. | Studies conducted before January 1st, 2013 or after December 31st, 2023. |
| **Study design** | - Primary studies providing original research findings in peer-reviewed journals. <br> - Observational studies (cross-sectional, cohort, case-control) in peer-reviewed journals. <br> - Review articles (systematic reviews, meta-analyses, narrative reviews) in peer-reviewed journals. <br> - Editorials, opinion pieces | Non-peer-reviewed articles, theses, dissertations, book reviews, news articles, conference reviews, or reports. |
| **Subject** | Studies evaluating or discussing access to healthcare for work related diseases and injuries. | Studies without specific reference to access to healthcare for work related diseases and injuries. |
| **Accessibility** | Studies or reports where full text is available. | Studies or reports where no full text is available. |

framed by our research question. Following previous recommendations [35], we did not include study outcome (access to healthcare) terms in search strategy to optimize the search results. The search strategy developed for each of the three databases is provided in S2 Appendix.

## Screening

The screening process will be performed in two stages: 1) an initial screening of titles and abstracts, followed by 2) an full-text screening. Five authors from the research team will conduct the screening. Four authors will be divided into two groups, with two reviewers assigned to each stage. Screening will be independently conducted to identify articles that meet the eligibility criteria. A senior reviewer will serve as the third reviewer to resolve discrepancies and crosscheck results between the two independent reviewers for each stage.

Inter-rater reliability tests will be conducted separately for each stage. Initially, two reviewers will independently assess a sample subset of 20% of studies to align on evaluation criteria and ensure a consistent inter-rater reliability rate of at least 85%. Multiple rounds of inter-rater reliability tests will be held until the expected 85% rate is achieved. After this alignment, the two reviewers will proceed to independently evaluate the remaining articles. Disagreements will be resolved by the senior reviewer to maintain objectivity in study selection. Reasons for excluding each article will be clearly recorded throughout the screening process. The selection process and decisions will be recorded in an Excel spreadsheet for detailed data extraction and analysis. At the time of writing, the literature search for this systematic review has not yet been completed.

## Data extraction

For each included study, data will be extracted across five domains:

a. Study characteristics (author, year, country, language, study design, sample size),

b. Population demographics (age, gender, origin nationality),

c. Exposures described (Migrant status, work-related disease or injury),

d. Outcomes measured (access to healthcare, timeliness, barriers, facilitators)

e. Key findings (effect sizes, thematic insights).

At least two reviewers will extract data to ensure the accuracy and reliability of the information collected. Discrepancies in data extraction between reviewers will be resolved through discussion, or by involving the third reviewer if necessary to achieve consensus. For missing data or unclear details, we will attempt to contact study authors for clarification or additional information. If authors cannot be reached, missing data will be recorded as "Not applicable". Extracted data will be organized and managed using an Excel spreadsheet.

### Risk of bias and quality assessment

In the article appraisal phase, the *Mixed Methods Appraisal Tool (MMAT)* [36] will be utilized to assess the quality of qualitative, quantitative, and mixed methods research. Each included study will be evaluated for the suitability of its design in addressing our research question, the adequacy of the sample size, participants recruitment bias, outcome measurement accuracy, and the clarity of result reporting. This meta-narrative approach allows for a systematic interpretation and integration of findings across diverse study designs.

For observational studies, the *ROBINS-I (Risk of Bias in Non-randomized Studies of Interventions)* tool [37] will be used to assess biases related to confounding, participants selection, and intervention classification.

### Data synthesis

Considering the expected heterogeneity among our included studies, we will primarily employ a narrative synthesis approach. This approach is chosen due to the anticipated diversity in study designs (qualitative, quantitative, mixed-methods), exposures (migrant status, work-related diseases and injuries), and outcomes (access to healthcare access, barriers, facilitators). Data will be synthesized when at least two studies report similar outcomes, ensuring our conclusions are drawn from robust evidence. Consistency across studies will be assessed based on the direction of effects and the contexts of interventions to determine the reliability of synthesized findings. We will synthesize findings related to primary outcomes, including healthcare access levels, barriers and facilitators to access. We will systematically categorize and compare studies, identifying common themes and differences.

While our primary focus will be on narrative synthesis, if we identify more than two quantitative studies that meet criteria for homogeneity, a meta-analysis will be considered to further quantify the effects and outcomes reported. Statistical methods such as meta-regression may be utilized to explore potential sources of heterogeneity and to provide a better understanding of the quantitative data. Additionally, sensitivity analysis may be employed to test the robustness of the findings, ensuring that our synthesis reflects the most reliable evidence available.

### Discussion

Our proposed systematic review has the potential to significantly influence how the GCC countries address access to healthcare for international labor migrants. One of our strengths is its comprehensive inclusion of the study population. We not only investigate migrant workers but also include expatriate professionals, providing an inclusive and culturally tailored perspective for the GCC countries. Our methodological rigor is enhanced by applying the gold standard of screening, where at least two reviewers independently screen the literature in parallel, rather than splitting it between reviewers to accelerate the process. This approach strengthens the robustness and reliability of our data collection. By recognizing and analyzing specific barriers and facilitators, our review will equip policymakers with the insights needed to promote more targeted interventions to improve timely access to healthcare among international labor migrants. Additionally, our review

will report work-related injuries across various sectors, which is vital for developing equitable healthcare policies that cater to all international labor migrants, regardless of their job type or industry. As a result, our findings can inform the development of evidence-based policies and healthcare interventions that are specifically designed to enhance access to healthcare services for international labor migrants in the GCC countries.

While our review is designed to be comprehensive and rigorous, there are some limitations that should be acknowledged. The variability in healthcare systems and migrant policies across the GCC countries presents a challenge in standardizing data collection and analysis. However, we will address this by employing robust data synthesis techniques and possibly sensitivity analysis to ensure our findings are as reliable as possible. Another potential limitation is the availability of data. Some information on migrant health may not be publicly available due to political sensitivities. To mitigate this, we will rely on peer-reviewed literature to ensure the credibility of our data. Moreover, while our team includes coauthors (A.D. and N.S.) proficient in Arabic, we limited our inclusion to English and Arabic studies that provide English abstracts. This decision was made to ensure the global relevance and accessibility of our findings, and to standardize the initial screening process. As a result, potentially relevant studies published in Arabic without English abstracts will be excluded, which may introduce a language-related limitation. Nonetheless, together with our efforts to contact the authors of included studies for missing data, we will do our best to minimize the impact of data availability issues. Lastly, another limitation is that we allowed data synthesis when at least two studies reported sufficiently similar outcomes. While syntheses based on three or more studies are considered more robust, we adopted a lower threshold to avoid excluding relevant evidence in our relatively understudied area.

Our final systematic review findings will be submitted to high-impact peer-reviewed journals for publication to ensure broad visibility. We will also present our findings at international seminars, conferences, and stakeholders' meetings to reach a wide audience and stimulate further discussion within professional communities.

## Supporting information

**S1 Appendix.  PRISMA-P 2015 checklist.**
(DOCX)

**S2 Appendix.  Search strategy for PubMed, Embase, and CINAHL.**
(DOCX)

## Acknowledgments

We extend our gratitude to Ms. Shannon Mulligan at Georgetown University Center for Global Health Practice and Impact, for her coordination and recruitment of collaborative authors from multidisciplinary backgrounds, which greatly enriched this project.

## Author contributions

**Conceptualization:** Weijun Yu, Katherine O. Robsky, Oliver Johnson, Bryan Shaw, Deus Bazira.

**Data curation:** Weijun Yu, Aleena Dawer, Jeanetta Floyd, Jiaqin Wu, Yulia Hutsul.

**Funding acquisition:** Deus Bazira.

**Investigation:** Weijun Yu, Aleena Dawer, Jeanetta Floyd, Deus Bazira.

**Methodology:** Weijun Yu, Jeanetta Floyd, Katherine O. Robsky, Oliver Johnson, Bryan Shaw, Deus Bazira.

**Project administration:** Weijun Yu, Deus Bazira.

**Resources:** Weijun Yu, Oliver Johnson, Deus Bazira.

**Supervision:** Weijun Yu, Deus Bazira.

**Validation:** Weijun Yu.

**Visualization:** Weijun Yu, Nicole Saad, Yulia Hutsul.

**Writing – original draft:** Weijun Yu, Aleena Dawer, Jeanetta Floyd, Nicole Saad, Yulia Hutsul, Dylan Ratnarajah, Deus Bazira.

**Writing – review & editing:** Weijun Yu, Nicole Saad, Jiaqin Wu, Yulia Hutsul, Dylan Ratnarajah, Martine Etienne-Mesubi, Deus Bazira.

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
