## [Decision Letter · Decision Letter 0]

Dear Dr. Yu,

Indicate which changes you require for acceptance versus which changes you recommendAddress any conflicts between the reviews so that it's clear which advice the authors should followProvide specific feedback from your evaluation of the manuscript

publication criteria  and not, for example, on novelty or perceived impact.

We look forward to receiving your revised manuscript.

Kind regards,

Sharada P Wasti, PhD

Academic Editor

PLOS ONE

Journal Requirements:

Additional Editor Comments:

Thank you for submitting your manuscript to the Journal of PLOS One. After careful review, we believe your work has strong potential but requires revision before it can be considered further. Our reviewers have provided constructive feedback to strengthen your manuscript. We invite you to submit a revised version addressing all points, providing a response letter detailing your revisions. We look forward to receiving your revised submission and appreciate your contributions to migrant health issues.

Reviewers' comments:

Reviewer's Responses to Questions

**Comments to the Author**

1. Does the manuscript provide a valid rationale for the proposed study, with clearly identified and justified research questions?

Reviewer #1: Partly

Reviewer #2: Yes

2. Is the protocol technically sound and planned in a manner that will lead to a meaningful outcome and allow testing the stated hypotheses?

Reviewer #1: Yes

Reviewer #2: Partly

3. Is the methodology feasible and described in sufficient detail to allow the work to be replicable?

Reviewer #1: Yes

Reviewer #2: Yes

4. Have the authors described where all data underlying the findings will be made available when the study is complete?

Reviewer #1: Yes

Reviewer #2: Yes

5. Is the manuscript presented in an intelligible fashion and written in standard English?

Reviewer #1: Yes

Reviewer #2: Yes

You may also provide optional suggestions and comments to authors that they might find helpful in planning their study.

Reviewer #1: Dear Authors,

I hope you find the following comments of use to you.

Introduction

Ln 93-96: A biased statement is made by linking the definition of international migrant workers and its contribution particularly to the Arab States. Migrant workers contribute to the economy in all counties that they work in. Give a general statement about this and remove the biasness.

Format intext citations when having more than one reference for a section. See lines 97-98, Ln 105, and Ln 114, for example. The same error is found thought the document.

Uncapitalize the first letter in International throughout the document unless it comes at the beginning of a sentence.

Ln 117-118: “.... nearly 30 million migrants [16], which accounts for more than 11.7% of the global migrant workforce [5]”: The 11.7% has no basis as the citation, reference [5], discusses output of research on the health of international migrant workers and is not related to the percentage given in this sentence.

Ln126: armature use of secondary sources of information. This statement is not substantiated “mortality of 6,500 migrant workers during the 2022 FIFA World Cup [18]”.

Ln 154-157: repetition of Ln 160-165. Consolidate the sections.

Methods

Study design

The authors are proposing to conduct the protocol based on PRISM-P 2015 and to report the full systematic review findings using PRISM 2020 guidelines. What are the rationales or such approach? Why not use PRISMA 2020 guidelines and checklists throughout?

Table 1: Time Frame: the exclusion criteria indicated studies conducted before January 1st, 2013. Does this mean studies conducted in 2014 will be included, while the eligibility criteria in Ln 203 explicitly indicates a time frame up to December 31st, 2023?

Table 1: Study design: The nature of the subject of the review might not be related to studies with RCT designs. Have the authors conducted a trials search for all the study types? Did they find any RCT on access to healthcare, barriers and facilitators to access? The matter will depend on a number of interrelated regulations and resources within each of the GCC countries. RCT might not be applicable to the subject of the study.

The authors did not address how they will handle literature published in Arabic. Are they only going to relay on the English abstract?

Data synthesis

Ln278-280: Explains the rationality of setting the minimum criteria at two studies reporting similar outcomes to synthesize your data. Why not at least three? Two would increase the likelihood of changes relative to three.

Discussion

Ln 310-321: repetition of the methods.

Funding Ln 349-350: be specific about the author sponsored by summer fellowship program by mentioning the order of in the list of authors.

Refences

Ref 24: incomplete

S1 Appendix:

Support: see funding above

Objectives: applied the PEO in place of PICO

Reviewer #2: Dear Authors,

I would like to commend the authors for their clear and structured approach to conducting this systematic review. The study is highly relevant and timely, addressing critical issues related to access to healthcare for work-related diseases and injuries among international migrants in the Gulf Cooperation Council countries. The proposed methodology is sound, with well-defined criteria for inclusion, clear stages of screening, and a thoughtful approach to data synthesis.

Overall, the manuscript is well-organized, but there are some areas where minor revisions in language, clarity, and formatting would enhance readability and ensure consistency with academic writing standards. The points raised in this review aim to strengthen the rigor of the study and improve the clarity of presentation. I believe that with these revisions, the manuscript will be in a strong position for publication.

Abstract Section:

Comment 1:

There are some language, clarity, and formatting issues that should be corrected for the abstract to meet academic standards.

For example:

• Original:

"International labor migrants are crucial to the global workforce in the Gulf Cooperation Council (GCC) countries, which host over 11.7% of the world’s migrant workforce, posing significant healthcare challenges."

Suggested Revision:

"International labor migrants form a significant part of the global workforce, particularly in the Gulf Cooperation Council (GCC) countries, which host over 11.7% of the world’s migrant labor force. This high concentration presents unique challenges in healthcare access and delivery."

• Original:

"Studies meeting eligibility criteria and focus on our outcome..."

Suggested Revision:

"Studies meeting the eligibility criteria and focusing on our primary outcome—access to healthcare—will be included."

Comment 2:

There is a small grammatical mistake in the following sentence:

"...to improve healthcare access for international migrants' in GCC countries."

Suggested Correction:

"...to improve healthcare access for international migrants in GCC countries."

Comment 3:

Abbreviations should be kept to a minimum in abstracts, as abstracts are often read independently. Using too many abbreviations can confuse readers. However, the abbreviation "GCC" is acceptable in this context because it is clearly defined at first mention and used consistently throughout the abstract.

Introduction section:

Recommendations:

1-Conduct a grammar and style check throughout the text.

2- Ensure terminology is consistent (e.g., migrant workers, expatriate professionals, international labor migrants).

3- Improve sentence clarity and transitions to strengthen the logical flow.

4- Review citation numbers to confirm they match up with references.

5- Maintain neutral and academic tone in discussing controversial topics like the kafala system.

Suggested Revision: Research questions

Additionally, the review will explore the following subtopics:

a. Barriers and facilitators affecting timely access to healthcare services

b. Policies developed and implemented to address constraints on healthcare access for international migrants

c. Scope and coverage of healthcare services available to this population

d. Affordability of services and healthcare financing mechanisms

e. Patterns of healthcare service utilization

f. Health workforce compatibility, including language barriers, cultural competence, and the availability of migrant-friendly services

g. Effectiveness of healthcare services, including quality, responsiveness, and overall impact

Method section:

The section is methodologically appropriate and well-structured. Minor revisions in phrasing and formatting would enhance clarity and professionalism.

Grammar & Syntax: Some sentence structures are too informal or awkward for academic writing (e.g., "Our PRISMA-P checklist is available in S1 Appendix" is better as "which is provided in S1 Appendix").

Consistency: Be consistent with naming of organizations and frameworks. Example: PRISMA-P vs PRISMA – always italicize or use quotation marks consistently if required by the journal.

Clarity: The sentence: "This study is determined not to involve human subjects..." could be clearer and more formal as "This study does not involve human participants..."

Tense Usage: Future tense “will be created” is fine, but using present perfect (“has been created”) may sometimes sound more formal, depending on the section and timing.

Academic Tone: Passive voice is often more suitable for formal tone (e.g., “was developed,” “were formally waived”).

Strengths:

• The step-by-step explanation of the screening workflow and quality control measures (e.g., inter-rater reliability testing, documentation of exclusions) enhances the methodological rigor.

• The use of a senior reviewer for conflict resolution is a good practice that supports objectivity.

**Do you want your identity to be public for this peer review?** For information about this choice, including consent withdrawal, please see our Privacy Policy

Reviewer #1: No

Reviewer #2: **Yes: ** Assist.Prof.Dr.Warda Hassan Abdullah

---

## [Author Response · Author response to Decision Letter 1]

16 May 2025

Dear Dr. Sharada P Wasti and reviewers,

We would like to sincerely thank you and the two reviewers for your thoughtful and detailed feedback on our manuscript. We deeply appreciate the time, effort, and expertise that you all dedicated to reviewing our work and providing such generous editorial and constructive comments.

Thanks to your valuable input, we have made significant improvements to the manuscript. Please find enclosed our point-by-point rebuttal letter, which carefully addresses each of the comments raised by you and the reviewers. As requested by PLOS ONE, we have also included both a track-changed version and a clean version of the revised manuscript.

---

## [Editor Report · Decision Letter 1]

Opportunities and challenges in access to healthcare for international migrants with work-related diseases and injuries in Gulf Cooperation Council countries: A systematic literature review protocol

PONE-D-24-25911R1

Dear Weijun Yu,

I greatly appreciate all authors for thoroughly reviewing your manuscript and addressing all the suggestions, which have been addressed well. We’re pleased to inform you that your manuscript has been judged scientifically suitable for publication and formally accepted for publication once it meets all outstanding technical requirements. We appreciate the effort you have put into addressing the reviewers' comments during the revision process, which has greatly enhanced the quality of your manuscript.

Within one week, you’ll receive an e-mail detailing the required amendments. When these have been addressed, you’ll receive a formal acceptance letter, and your manuscript will be scheduled for publication.

If your institution or institutions have a press office, please notify them about your upcoming paper to help maximise its impact. If they’ll be preparing press materials, please inform our press team as soon as possible – no later than 48 hours after receiving the formal acceptance. Your manuscript will remain under strict press embargo until 2 pm Eastern Time on the date of publication. For more information, please contact onepress@plos.org.

Thank you for choosing the Journal of PLOS ONE for disseminating your research. We look forward to seeing your work published and hope it will inspire further advancements in the field.

Kind regards,

Sharada P Wasti, MSc, PhD

Academic Editor

PLOS ONE

---

## [Editor Report · Acceptance letter]

PONE-D-24-25911R1

PLOS ONE

Dear Dr. Yu,

I'm pleased to inform you that your manuscript has been deemed suitable for publication in PLOS ONE. Congratulations! Your manuscript is now being handed over to our production team.

Kind regards,

on behalf of

Dr. Sharada P Wasti

Academic Editor

PLOS ONE